# Physical well-being recovery trajectories by reconstruction modality in women undergoing mastectomy and breast reconstruction: Significant predictors and health-related quality of life outcomes

Cai Xu[1,2]*, Peiyi Lu[3], André Pfob[2,4], Andrea L. Pusic[5], Jennifer B. Hamill[6], Chris Sidey-Gibbons[1,2]

1 Division of Internal Medicine, Section of Patient Centered Analytics, The University of Texas MD Anderson Cancer Center, Houston, TX, United States of America, 2 MD Anderson Center for INSPiRED Cancer Care (Integrated Systems for Patient-Reported Data), The University of Texas MD Anderson Cancer Center, Houston, TX, United States of America, 3 Department of Epidemiology, Mailman School of Public Health, Columbia University, New York, NY, United States of America, 4 Department of Obstetrics & Gynecology, Heidelberg University Hospital, Heidelberg, Germany, 5 Department of Surgery, Patient-Reported Outcome Value & Experience (PROVE) Center, Harvard Medical School & Brigham and Women's Hospital, Boston, MA, United States of America, 6 The Department of Surgery, Section of Plastic and Reconstructive Surgery, University of Michigan, Ann Arbor, MI, United States of America

* Cairxu@gmail.com

## Abstract

### Objectives

We sought to identify trajectories of patient-reported outcomes, specifically physical well-being of the chest (PWBC), in patients who underwent postmastectomy breast reconstruction, and further assessed its significant predictors, and its relationship with health-related quality of life (HRQOL).

### Methods

We used data collected as part of the Mastectomy Reconstruction Outcomes Consortium study within a 2-year follow-up in 2012–2017, with 1422, 1218,1199, and 1417 repeated measures at assessment timepoints of 0,3,12, and 24 months, respectively. We performed latent class growth analysis (LCGA) in the implant group (IMPG) and autologous group (AUTOG) to identify longitudinal change trajectories, and then assessed its significant predictors, and its relationship with HRQOL by conducting multinomial logistic regression.

### Results

Of the included 1424 patients, 843 were in IMPG, and 581 were in AUTOG. Both groups experienced reduced PWBC at follow-up. LCGA identified four distinct PWBC trajectories ($\chi2 = 1019.91$, $p<0.001$): low vs medium high vs medium low vs high baseline PWBC that was restored vs. not-restored after 2 years. In 76.63%(n = 646) of patients in IMPG and

**Data Availability Statement:** All relevant data are within the manuscript and its Supporting Information files as S1 Data.

**Funding:** This study is supported by National Cancer Institute Grant No. R01 CA152192 and in part by National Cancer Institute Support Grant No. P30 CA008748, but the funders had no role in study design, data collection and analysis, decision to publish, or preparation of the manuscript.

**Competing interests:** Cai Xu No relationship to disclose. Peiyi Lu No relationship to disclose. André Pfob No relationship to disclose. Andrea L. Pusic Patents, Royalties, Other Intellectual Property: I am a codeveloper of BREAST-Q and receive royalty payments when it is used in for-profit industry-sponsored trials. Jennifer B Hamill No relationship to disclose. Chris Sidey-Gibbons No relationship to disclose. Our competing interests statement does not alter our adherence to PLOS ONE policies on sharing data and materials.

**Abbreviations:** PRO, patient-reported outcome; HRQOL, health-related quality of life; MCID, minimal clinically important differences; IMPG, implant group; AUTOG, autologous group; PWBC, physical well-being of the chest; PMBR, postmastectomy breast reconstruction.

62.99% (n = 366) in AUTOG, PWBC was restored after two years. Patients in IMPG exhibited worse PWBC at 3 months post-surgery than that in AUTOG. Patients with low baseline PWBC that did not improve at 2-year follow up (n = 28, 4.82% for AUTOG) were characterized by radiation following reconstruction and non-white ethnicity. In IMPG, patients with medium low-restored trajectory were more likely to experience improved breast satisfaction, while patients developing high-restored trajectories were less likely to have worsened psychosocial well-being.

## Conclusion

Although more women in IMPG experienced restored PWBC after 2 years, those in AUTOG exhibited a more favorable postoperative trajectory of change in PWBC. This finding can inform clinical treatment decisions, help manage patient expectations for recovery, and develop rehabilitation interventions contributing to enhancing the postoperative quality of life for breast cancer patients.

## Introduction

Health-related quality of life (HRQOL) following postmastectomy breast reconstruction (PMBR) for breast cancer patients has experienced an increase in interest as the number of breast reconstructions and bilateral mastectomies rises [1]. The Breast-Q questionnaire, as a validated and reliable patient-reported outcome (PRO) measurement developed for breast surgery [2], can be utilized to evaluate HRQOL across multiple domains, including breast satisfaction, physical well-being of the chest (PWBC), physical well-being of the abdomen, sexual well-being, and psychosocial well-being [3,4]. Women who receive PMBR demonstrate better breast-related body image compared to those who receive a mastectomy without reconstruction [3,5].

Cancer patients often experience functional deficits following treatment, which can limit their physical capacity [6]. Previous studies examining long-term effects of upper limb dysfunction show more than half of patients undergoing breast cancer surgery had upper quadrant dysfunction up to 6 years postdiagnosis [7]. Similarly, persistent functional deficits affected a large proportion of women at 1.5 years postoperatively [8]. Women's PWBC was restored 2 years after experiencing significantly worsened PWBC after 1 year, but group differences by modality have not investigated yet [9]. Significantly, improved PWBC was reported by patients indicated that patients benefit more from either implant-based or autologous construction than no reconstruction [10]. Notably, autologous reconstruction was superior to implant-based reconstruction in significantly reducing chest and upper body morbidity [11]. Nevertheless, after studying the impact of PMBR on HRQOL by modality 1 year after reconstruction, other researchers have come to very different conclusions, reporting that neither the implant nor autologous group had recovered to the baseline function in their chest [12]. A systematic review of reconstruction modalities shows that no demonstrable overall differences in PWBC were found between the implant and autologous reconstructions [13].

The functional impact of PMBR among different reconstruction types has been assessed in several studies as a key outcome [14]. However, to date, the recovery trajectories of the PWBC in long follow-up periods between patients within different reconstruct modalities and their association with other HRQOL outcomes have not been systematically investigated. We

sought to fill this knowledge gap using data collected in a multicenter, prospective study of women who underwent PMBR and were followed for up to 2 years. We aimed to provide definitive comparisons of reconstruction types using functional PRO, in order to better understand inconsistent findings reported in previous studies [15].

We hypothesize that distinct growth trajectories in PWBC exist for breast cancer patients who have received either the implant group (IMPG) or autologous group (AUTOG), and that these growth trajectories are associated with PROs in breast satisfaction, sexual well-being, and psychosocial well-being. We sought to account for the heterogeneity between breast cancer patients in change of PWBC under different reconstruction procedures, by identifying latent subgroups with distinct growth trajectories over time. These data-driven findings on physical function recovery trajectories have the potential to inform clinical decision-making to help patients achieve desired health outcomes after reconstruction. Furthermore, the identified high-risk/low-risk predictors may provide insights for the development of innovative rehabilitation interventions to facilitate patient-centered and goal-concordant care and ultimately improve patients' postoperative quality of life in clinical practice.

## Methods

### Data and sample

Data collected at 11 study sites across the United States and Canada from 2012 to 2017 in an international multicenter trial (Mastectomy Reconstruction Outcomes Consortium (MROC) study, NCT01723423) were used [4]. This trial focused on women who underwent PMBR and assessed their HRQOL in varied time intervals within the 2 years post-surgery. The MROC study recruited patients aged 18+, undergoing bilateral or unilateral, immediate, or delayed PMBR with the goal of risk reducing or therapeutic, and excluded patients with previous failed attempts of PMBR. This study was approved at all include centers by the corresponding institutional review board (IRB) or research ethics board (REB) depending on country. Written informed consent was obtained from all participants prior to enrollment. No one under the age of 18 years was approached or enrolled.

To investigate PWBC trajectories following PMBR as well as its association with change in breast satisfaction, sexual well-being, and psychosocial well-being, 4 waves of PRO assessed at baseline, 3-month, 1-year, and 2-year follow-up were used. Furthermore, patients included in this study must provide at least a one-time point of observation for PWBC and were treated with either implant-based or autologous reconstruction. Patients without baseline or 2-year follow-up scores on either of these 3 associated HRQOL domains were excluded.

### Sociodemographic variables

Age and BMI were continuous. Race (White/non-white), diabetes(yes/no), smoke status(yes/no), and simplified marital status (partnerless/partnered) were binary. Simplified educational level was grouped into two levels: high school degree and below versus above high school degree. Simplified working status was categorized into two types: employed and others. Household income per year was ordinal with a middle-income range of $50,000 to $99,999.

### Clinical and patient-reported covariates

Pre-operative PRO data were comprised of satisfaction with breast, PWBC, physical well-being of abdomen, psychosocial, and sexual well-being. The follow-up PRO data of PWBC was assessed at 3, 12, and 24 months after surgery, while the breast satisfaction, psychosocial, and sexual well-beings were all evaluated at 2 years. Radiation refers to the patients who received

this therapy before or after reconstruction with 3 levels (before/after/none). The mastectomy type (simple/nipple-sparing), chemotherapy (received/not received), reconstruction laterality (unilateral/ bilateral), and mastectomy indication (therapeutic/prophylactic) were coded as binary variables. Axillary intervention type was measured using 3 types: none, sentinel lymph node biopsy (SLNB), and axillary lymph node dissection (ALND).

## Health-related quality of life outcome

The HRQOL after PMBR was measured using the Breast-Q instrument. Here, 'Satisfaction with Breasts,' 'Psychosocial Well-being,' 'Sexual Well-Being,' 'Physical Well-Being: Chest,' and 'Physical Well-Being: Abdomen' subscales were adopted. Each independent subscale was rated with a converted score ranging from 0 (worst) to 100 (best). Previous research identified minimal clinically important differences (MCID) at a score of 4 for breast satisfaction, sexual well-being, and psychosocial well-being domains in patients with reconstructed breasts [16]. Hence, we defined 3 types of outcomes for each domain, respectively, by comparing its corresponding PRO at baseline and 2-year follow-up. Specifically, 1) if the change was equal or greater than the positive MCID of 4, improved; 2) if the change was equal or less than the negative MCID of 4, worsened; 3) otherwise, it stable.

All these variables (See S1 Table 1 in S1 File) that were pre-assessed and proved free of multicollinearity concerns were included in the final analysis.

## Analytic strategy

We stratified data into two groups based on reconstruction modality:1) IMPG and 2) AUTOG. A proper Chi-square test or T-test was conducted to assess the differences between them. Then, latent class growth analysis (LCGA) was performed on them, respectively. LCGA used here was to identify longitudinal changes and classify individuals into different latent subgroups based on their common growth PWBC trajectories [17]. LCGA chosen was attributed to the special characteristics that it estimates the average growth in the longitudinal data, featured by fixed intercept and slope per class, facilitating the interpretation and estimation. Additionally, LCGA has used full information maximum likelihood (FIML) to deal with missing data, which has been demonstrated more efficient and accurate than the imputation or list-wise deletion approach [18].

We first built the unconditional growth model of PWBC trajectories including time only as a covariate to generate initial start values, and then continually estimated this model with a pre-specified gradually increasing number of classes to explore the best one [19]. The optimal model to be selected should satisfy certain commonly used goodness-of-fit indices, including three indices of Bayesian Information criterion (BIC), Akaike Information Criterion (AIC), and sample-size adjusted BIC (SSBIC), and entropy values, among which, BIC is recommended as the most reliable [20]. Meanwhile, the Lo-Mendell-Rubin likelihood ratio test (LMR LRT) was performed to assess whether models with more classes were statistically significantly better than the model with fewer classes. The chosen model should have smaller values for the information criterion indices, higher entropy score, and significant $p$-value for LMR LRT, and also take into account model interpretability and parsimony factors [21].

After finalizing the optimal number of latent PWBC trajectories, each included patient will be assigned a new class membership based on their respective posterior class probabilities to represent her growth patterns over time, and their respective new class membership attribution will not change at all the assessment time points. Then, multinomial logistic regression was conducted to examine significate indicators for predicting the new class memberships. Finally, we also conducted a series of multinomial logistic regression models to examine the

associations of PWBC trajectories with the changes in breast satisfaction, sexual well-being, and psychosocial well-being within the 2 years after PMBR using 3 pre-defined outcomes: improved, stable, and worsened.

All the analyses were performed using the R-4.2.1 software with packages "lcmm" [22], "nnet", "ggplot2".

## Inclusivity in global research

Additional information regarding the ethical, cultural, and scientific considerations specific to inclusivity in global research is included in the (S1 Data).

# Results

## Sample characteristics

Of the 1424 patients included, 843 were in IMPG, and 581 were in AUTOG. Table 1 shows that AUTOG had a significantly higher average age (51.74 vs 48.55) and BMI (28.61 vs 24.96), but significantly lower PRO at baseline for all Breast-Q scales; For the follow-up PRO assessment, the IMPG had significantly higher PWBC at 24 months (77.26 vs 75.13) and significantly lower PWBC at 3 months (69.42 vs 71.76), sexual well-being at 24 months (53.50 vs 55.92), and breast satisfaction at 24 months (63.52 vs 67.37). Furthermore, statistically, significant differences were also observed among all the included categorical variables except smoke, chemotherapy, marital status, working status, and race.

## Physical well-being of chest trajectories: LCGA results

Fit statistics in Table 2 reasonably supported the selection of the 4-class model, which has the smallest BIC, as a final model for both IMPG and AUTOG. Based on the predictive mean and sample mean of PWBC values for each subgroup at all assessment time points (see S1 Table 2 in S1 File), Fig 1 plots the distinct 4 PWBC trajectories with recoded group names according to their predicted baseline function level (low, medium-low, medium-high, high) and final recovery result (restored, not restored). These trajectories denoted the temporal trends for groups of individuals with more homogeneity in the parameters.

In IMPG, 646(76.63%) of patients had fully restored PWBC, of which 380(58.82%) were with medium low-restored trajectory, and 266(41.18%) were with high-restored trajectory. Of these 197(23.37%) patients with PWBC not returning to baseline function, 87 patients (44.16%) with medium-high baseline levels experienced a sharp decline within the early 3 months after surgery. In AUTOG, the largest class was patients (n = 261, 44.92%) with medium-high restored trajectory; the smallest class was comprised of 28 patients (4.82%) with low-not restored trajectory.

Results of the Chi-square test indicated that these trajectories between IMPG and AUTOG were statistically significant ($\chi2$ = 1019.91, $p<0.001$). Patients in AUTOG were more likely to have medium high-restored(residual = 14.97) and medium-low-not restored (residual = 12.67) trajectories, whereas patients in IMPG were more likely to develop medium-low restored (residual = 10.34) health outcomes based on residuals. Demographics for all patients in different trajectories were presented in S1 Tables 3 and 4 in S1 File.

## Significant predictors for new class membership

Results in Table 3 suggest that baseline PROs of PWBC and physical well-being of the abdomen were significantly, and strongly associated with predicting new class membership for both IMPG and AUTOG ($p<0.001$). For IMPG, the odds of developing high-restored and

**Table 1. Baseline demographic and clinical characteristics of participants for physical well-being scale.**

| | Full sample (n = 1424) | Implant-based reconstruction (n = 843) | Autologous reconstruction (n = 581) | p value[a] |
|---|---|---|---|---|
| **Patient variables** | | | | |
| Age, mean (SD), years | 49.85(9.90) | 48.55(10.31) | 51.74(8.94) | <0.001[b] |
| BMI, mean (SD), kg/m$^2$ | 26.45(5.37) | 24.96(4.90) | 28.61(5.30) | < 0.001[b] |
| Diabetes, no (%) | | | | <0.001[c] |
| No, no. (%) | 1361(95.58) | 819(97.15) | 542(93.29) | |
| Yes, no. (%) | 63(4.42) | 24(2.85) | 39(6.71) | |
| Smoker | | | | 0.918[c] |
| No, no. (%) | 1387(97.40) | 818(97.03) | 569(97.93) | |
| Yes, no. (%) | 25(1.76) | 15(1.78) | 10(1.72) | |
| Unknown, no. (%) | 12(0.84) | 10(1.19) | 2(0.34) | |
| **Pre-operative patient-reported outcome data** | | | | |
| BREAST-Q satisfaction with breast, mean (SD), 0–100 | 60.42 (22.13) | 64.02(22.14) | 55.20(21.06) | <0.001[b] |
| BREAST-Q physical well-being chest and upper body, mean (SD), 0–100 | 79.12 (14.37) | 80.78(13.76) | 76.73(14.89) | <0.001[b] |
| BREAST-Q psychosocial well-being, mean (SD), 0–100 | 69.65 (18.03) | 71.94(17.34) | 66.32(18.50) | <0.001[b] |
| BREAST-Q physical well-being abdomen, mean (SD), 0–100 | 89.58 (13.31) | 90.88(12.51) | 87.69(14.19) | <0.001[b] |
| BREAST-Q sexual well-being, mean (SD), 0–100 | 55.63 (20.39) | 59.33(18.97) | 50.27(21.20) | <0.001[b] |
| **Follow-up patient-reported outcome data** | | | | |
| BREAST-Q physical well-being chest and upper body at 3 months, mean (SD), 0–100 | 70.38 (13.59) | 69.42(13.03) | 71.76(14.25) | 0.003[b] |
| BREAST-Q physical well-being chest and upper body at 12 months, mean (SD), 0–100 | 75.26 (14.79) | 75.72(14.59) | 74.62(15.06) | 0.205[b] |
| BREAST-Q physical well-being chest and upper body at 24 months, mean (SD), 0–100 | 76.39 (14.99) | 77.26(14.39) | 75.13(15.74) | 0.010[b] |
| BREAST-Q psychosocial well-being at 24 months, mean (SD), 0–100 | 74.33 (19.13) | 74.00(19.19) | 74.80(19.05) | 0.443[b] |
| BREAST-Q sexual well-being at 24 months, mean (SD), 0–100 | 54.49 (21.97) | 53.50(21.44) | 55.92(22.67) | 0.043[b] |
| BREAST-Q breast satisfaction at 24 months, mean (SD), 0–100 | 65.09 (18.59) | 63.52(18.11) | 67.37(19.04) | <0.001[b] |
| Radiation | | | | |
| After reconstruction, no. (%) | 274(19.24) | 141(16.73) | 133(22.89) | 0.004[c] |
| Before reconstruction, no. (%) | 174(12.22) | 36(4.27) | 138(23.75) | <0.001[c] |
| None, no. (%) | 976(68.54) | 666(79.00) | 310(53.36) | <0.001[c] |
| Mastectomy | | | | <0.001[c] |
| Nipple-sparing, no. (%) | 165(11.59) | 153(18.15) | 12(2.07) | |
| Simple, no. (%) | 1259(88.41) | 690(81.85) | 569(97.93) | |
| Chemotherapy | | | | 0.239[c] |
| Received, no. (%) | 412(28.93) | 234(27.76) | 178(30.64) | |
| Not received, no. (%) | 1012(71.07) | 609(72.24) | 403(69.36) | |
| Reconstruction laterality | | | | <0.001[c] |
| Unilateral, no. (%) | 652(45.79) | 316(37.49) | 336(57.83) | |
| Bilateral, no. (%) | 772(54.21) | 527(62.51) | 245(42.17) | |
| Mastectomy indication | | | | 0.007[c] |

*(Continued)*

**Table 1.** (Continued)

| | Full sample (n = 1424) | Implant-based reconstruction (n = 843) | Autologous reconstruction (n = 581) | p value[a] |
|---|---|---|---|---|
| Therapeutic, no. (%) | 1282(90.03) | 744(88.26) | 538(92.6) | |
| Prophylactic, no. (%) | 142(9.97) | 99(11.74) | 43(7.4) | |
| Axillary intervention | | | | |
| Axillary lymph node dissection (ALND), no. (%) | 379(26.62) | 241(28.59) | 138(23.75) | **0.042[c]** |
| Sentinel lymph node biopsy (SLNB), no. (%) | 644(45.22) | 404(47.92) | 240(41.31) | **0.014[c]** |
| None, no. (%) | 401(28.16) | 198(23.49) | 203(34.94) | **<0.001** |
| **Socioeconomic and ethnic data** | | | | |
| Marital status | | | | 0.882[c] |
| Partnerless, no. (%) | 245(17.21) | 146(17.32) | 99(17.04) | |
| Partnered, no. (%) | 1173(82.37) | 693(82.21) | 480(82.62) | |
| Unknown, no. (%) | 6(0.42) | 4(0.47) | 2(0.34) | |
| Education level | | | | **<0.001[c]** |
| High school degree and below, no. (%) | 130(9.13) | 53(6.29) | 77(13.25) | |
| Above high school degree, no. (%) | 1292(90.73) | 789(93.59) | 503(86.57) | |
| Unknown, no. (%) | 2(0.14) | 1(0.12) | 1(0.17) | |
| Working status | | | | 0.404[c] |
| Others, no. (%) | 407(28.58) | 248(29.42) | 159(27.37) | |
| Employed, no. (%) | 1003(70.44) | 587(69.63) | 416(71.60) | |
| Unknown, no. (%) | 14(0.98) | 8(0.95) | 6(1.03) | |
| Household income per year | | | | |
| <50,000$, no. (%) | 211(14.82) | 97(11.51) | 114(19.62) | **<0.001[c]** |
| $50,000 to $99,999$, no. (%) | 460(32.30) | 226(26.81) | 234(40.28) | **<0.001[c]** |
| >$100,000$, no. (%) | 709(49.79) | 494(58.60) | 215(37.01) | **<0.001[c]** |
| Unknown, no. (%) | 44(3.09) | 26(3.08) | 18(3.10) | |
| Race | | | | 0.400[c] |
| White, no. (%) | 1290(90.59) | 768(91.10) | 522(89.85) | |
| Non-White, no. (%) | 124(8.71) | 69(8.19) | 55(9.47) | |
| Unknown, no. (%) | 10(0.70) | 6(0.71) | 4(0.69) | |

[a]P values refer to differences in the implant-based and autologous reconstruction groups. P values < 0.05 highlighted in bold.

[b]P values refer to t-tests to evaluate mean differences of continuous data.

[c]P values refer to Chi-square tests for binary variable evaluation (variable true vs. variable not true).

medium low-restored trajectories for patients without radiation therapy were 3.75 and 3.38 times as high as that for patients undergoing radiation after reconstruction, respectively. For AUTOG, patients without axillary intervention, undergoing chemotherapy, and having higher baseline PRO psychosocial well-being were more likely not to develop low-not restored trajectories due to all their risk ratio significantly greater than 1.

## Associated breast satisfaction, sexual, and psychosocial well-being change over time

As shown in condensed Table 4 and detailed S1 Table 5 in S1 File, for patients in IMPG, compared to the stable outcome of breast satisfaction at 2-year follow-up, the odds of experiencing improved breast satisfaction for patients with medium low-restored trajectory and SLNB intervention were 2.60 and 1.90 times as high as that for patients developing low-not restored

**Table 2. Model fit indices of latent class growth analysis models with different numbers of classes.**

| No. of classes | Log Likelihood | AIC | BIC | SABIC | ENTROPY | LMR LRT (*p* value) |
|---|---|---|---|---|---|---|
| **Implant-based reconstruction** | | | | | | |
| 1 | -12533.33 | 25076.67 | 25100.35 | 25084.48 | 1.0000000 | |
| 2 | -12264.22 | 24548.44 | 24595.81 | 24564.05 | 0.6508140 | < 0.001 |
| 3 | -12177.83 | 24385.67 | 24456.72 | 24409.09 | 0.7092395 | < 0.001 |
| 4 | -12156.44 | 24352.89 | 24447.63 | 24384.11 | 0.6583209 | < 0.001 |
| 5 | -12147.35 | 24344.70 | 24463.12 | 24383.73 | 0.5990494 | 0.004 |
| **Autologous reconstruction** | | | | | | |
| 1 | -8935.437 | 17880.87 | 17902.70 | 17886.82 | 1.0000000 | |
| 2 | -8663.407 | 17346.81 | 17390.46 | 17358.72 | 0.7373962 | < 0.001 |
| 3 | -8602.918 | 17235.84 | 17301.31 | 17253.69 | 0.7242673 | < 0.001 |
| 4 | -8577.173 | 17194.35 | 17281.64 | 17218.15 | 0.7033583 | < 0.001 |
| 5 | -8565.041 | 17180.08 | 17289.20 | 17209.84 | 0.6479228 | < 0.001 |

trajectory and with ALND intervention, respectively. S1 Table 6 in S1 File shows, in IMPG, undergoing chemotherapy, not smoking, being partnerless, and having higher baseline PRO breast scores were significantly associated with worsened sexual well-being based on their separate risk ratio significantly greater than 1. In AUTOG, a simple type of mastectomy and less than $50,000 were significantly associated with improved sexual well-being (*p*<0.05). Nevertheless, the odds of experiencing worsened sexual wellbeing for patients being partnerless was 2.62 times as highs as that for patients being partnered. Compared to the stable outcome of psychosocial well-being in S1 Table 7 in S1 File, in IMPG, patients being partnerless, were only 54% as likely as patients being partnered to have improved psychosocial well-being; when the baseline breast score increased by 1 unit, the odds of experiencing improved psychosocial well-being were only 99% as high. Hence, patients being partnerless, with higher baseline PRO breast scores were less likely to have improved psychosocial well-being. Patients developing high-restored trajectories were only 32% as likely as patients developing low-not restored trajectories to experience worsened psychosocial well-being with reconstructed breast. In AUTOG, patients without undergoing axillary intervention were 67% less likely than patients undergoing ALND intervention to have worsened psychosocial well-being.

## Discussion

### Main principal findings

Within 2 years after surgery, patients' PWBC was, on average, not fully recovered regardless of undergoing implant-based or autologous reconstruction [12]. The ascertained significant differences in all the physical well-being PROs between IMPG and AUTOG echo other researchers' findings that patients in IMPG had more severe issues in physical function, which may relate to the side effects or characteristics of implant reconstruction procedure [23,24]. We observe that although included patients in IMPG had significantly higher PRO than patients in AUTOG at baseline for all subscales, whereas patients in AUTOG did have significantly higher sexual well-being, and were more satisfied, on average, with their reconstructed breasts at 2 years after mastectomy, which is consistent with existing literature [12,25,26].

Within IMPG and AUTOG, substantial discrepancies in growth PWBC trajectory patterns were observed following breast reconstruction. This observation is consistent with the previous finding that during the 3 months early recovery period, the AUTOG experienced less

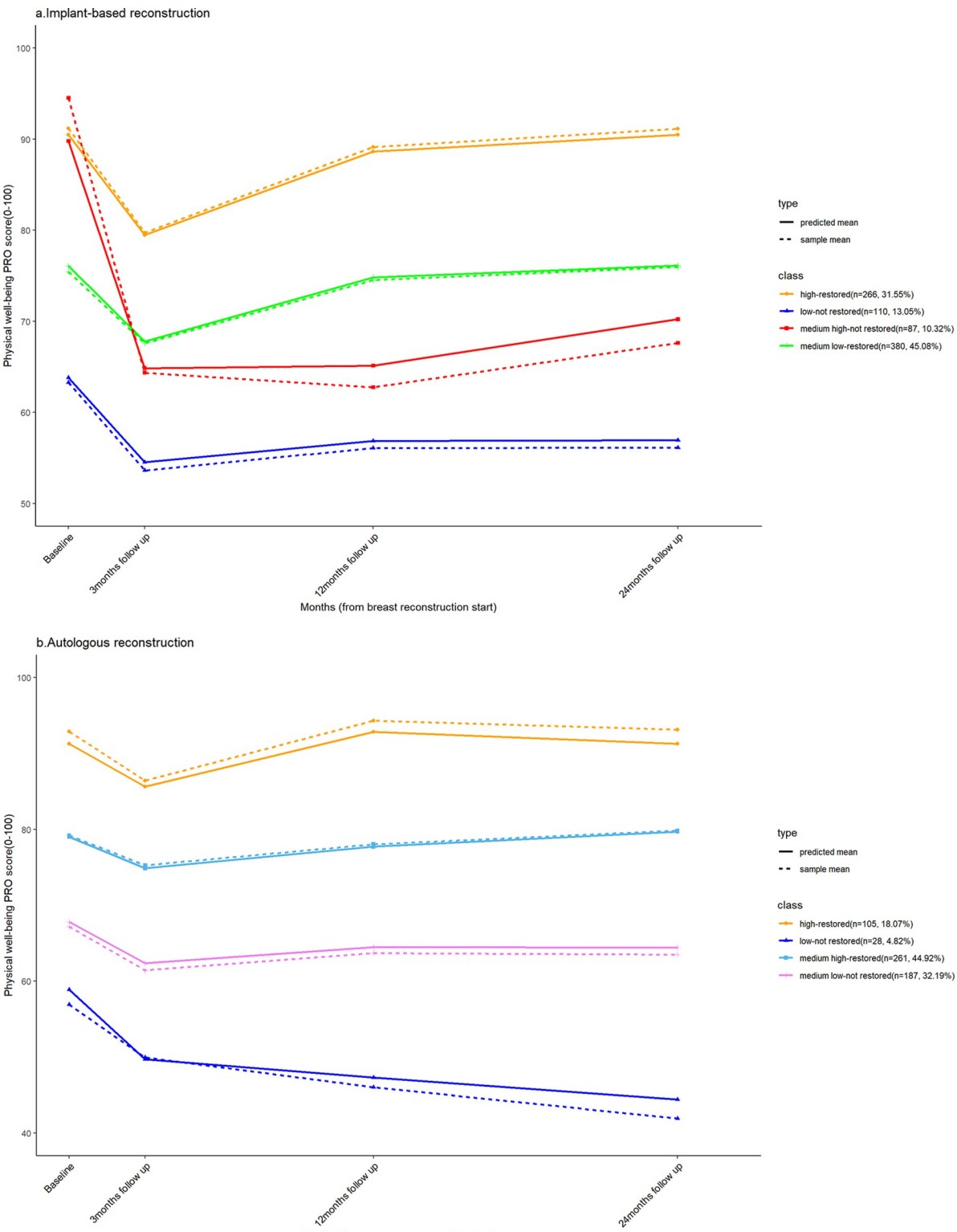

**Fig 1. Physical well-being trajectories following postmastectomy breast reconstruction by reconstruction modality.**

**Table 3. Multinomial logistic regression models predicting new class membership.**

|  | Implant-based reconstruction (n = 792) | | | Autologous reconstruction (n = 549) | | |
|---|---|---|---|---|---|---|
|  | high-restored vs low-not restored | medium high-not restored vs low-not restored | medium low-restored vs low-not restored | high-restored vs low-not restored | medium high-restored vs low-not restored | medium low-not restored vs low-not restored |
| Laterality: unilateral[a] | 1.39 | 1.65 | 1.21 | 1.03 | 0.69 | 0.82 |
| Indication: therapeutic[a] | 0.44 | 0.26 | 0.63 | 7.82 | 2.17 | 2.06 |
| Mastectomy: simple[a] | 0.84 | 0.78 | 0.63 | 0.00***[c] | 0.00***[c] | 0.00***[c] |
| Axillary[a] |  |  |  |  |  |  |
| none | 1.51 | 0.65 | 1.00 | 9.23* | 7.22* | 7.95* |
| SLNB | 2.25 | 1.74 | 1.01 | 5.87 | 6.14* | 5.91* |
| BMI[a] | 0.92* | 0.97 | 0.98 | 0.97 | 0.96 | 0.94 |
| Diabetes[a]: no | 0.11 | 0.06* | 0.28 | 0.95 | 0.78 | 0.40 |
| Radiation |  |  |  |  |  |  |
| before | 3.21 | 0.81 | 2.21 | 11.29 | 3.70 | 1.10 |
| none | 3.75* | 0.88 | 3.38** | 5.08 | 1.48 | 0.85 |
| Chemotherapy[a]: yes | 1.02 | 0.50 | 1.18 | 7.92* | 8.01** | 7.99** |
| Age[a] | 0.99 | 0.98 | 1.00 | 1.03 | 1.03 | 1.05 |
| Smoker[a,b]: no | 0.75 | 0.65 | 0.87 | 0.00*** | 0.09 | 0.30 |
| Marital[a]: partnerless | 0.92 | 0.98 | 1.20 | 0.43 | 0.40 | 0.34 |
| Education[a]: high school and below | 1.07 | 0.61 | 2.27 | 0.78 | 0.42 | 0.25 |
| Work[a]: others | 0.69 | 0.63 | 0.73 | 1.24 | 0.79 | 0.47 |
| Income[a] |  |  |  |  |  |  |
| $50,000-$99,999 | 1.31 | 1.52 | 1.47 | 1.85 | 1.84 | 2.25 |
| Less than $50,000 | 0.69 | 0.54 | 0.41* | 1.66 | 2.46 | 3.78 |
| Race[a]: white | 1.74 | 1.41 | 2.36 | 2.10 | 1.01 | 0.74 |
| Baseline PRO breast[a] | 1.00 | 0.99 | 1.00 | 0.99 | 0.99 | 0.98 |
| Baseline PRO psychosocial[a] | 1.01 | 1.01 | 1.01 | 1.06* | 1.05* | 1.05* |
| Baseline PRO physical[a] | 1.35*** | 1.42*** | 1.13*** | 1.36*** | 1.19*** | 1.07** |
| Baseline PRO physical abdomen[a] | 1.05** | 1.03 | 1.01 | 1.10*** | 1.05** | 1.04** |
| Baseline PRO sexual[a] | 1 | 1 | 1 | 0.96 | 0.97 | 0.97 |

[a]The independent variables were all measured at baseline.

[b]Reference group of smoker variable for patients with autologous reconstruction subgroup was "no".

[c]Coefficients were relative risk ratio. Risk ratios were approaching 0 for some variables due to the small sample size for that variable. Standard errors are robust.

*$p<0.05$

**$p<0.01$

***$p<0.001$.

chest and upper body physical morbidity than the IMPG although PWBC in either group had returned to baseline level [27]. The differences reflect that patients in IMPG had reported more symptoms affecting their physical function, while patients in AUTOG were more likely to report more symptoms in the abdomen [12]. The physical well-being score in AUTOG is significantly better than that in IMPG [11,28].

Of particular note, about 87(10.32%) of patients with predicted medium-high baseline PWBC in IMPG experienced the steepest declines and unrestored PWBC in the end. Results show those patients tended to have the highest baseline PWBC and baseline psychosocial well-

**Table 4. Multinomial logistic regression models predicting health-related quality of life outcome based on class membership of physical well-being trajectory.**

| Health-related quality of life | Implant-based reconstruction (n = 793) | | Autologous reconstruction (n = 549) | |
|---|---|---|---|---|
| | Improved vs stable[b] | Worsened vs stable[b] | Improved vs stable[b] | Worsened vs stable[b] |
| **Breast satisfaction outcome** | | | | |
| Class[a] | | | | |
| high-restored | 2.36 | 0.68 | 3.21 | 1.23 |
| medium high-not restored | 1.97 | 0.95 | | |
| medium low-restored | 2.60* | 1.04 | | |
| medium high-restored | | | 3.67 | 2.32 |
| medium low- not restored | | | 3.20 | 2.14 |
| **Sexual well-being outcome** | | | | |
| Class[a] | | | | |
| high-restored | 1.30 | 0.35* | 1.07 | 0.14* |
| medium high-not restored | 1.08 | 0.47 | | |
| medium low-restored | 1.36 | 0.56 | | |
| medium high-restored | | | 1.02 | 0.21 |
| medium low- not restored | | | 1.34 | 0.44 |
| **Psychosocial well-being outcome** | | | | |
| Class[a] | | | | |
| high-restored | 1.25 | 0.32** | 1.27 | 0.62 |
| medium high-not restored | 1.39 | 0.86 | | |
| medium low-restored | 1.00 | 0.51 | | |
| medium high-restored | | | 1.55 | 0.68 |
| medium low- not restored | | | 1.64 | 1.48 |

[a]The reference group for class membership is "low-not restored". Coefficients are relative risk ratio. Standard errors are robust.

$*p<0.05$

$**p<0.01$

$***p<0.001$.

[b]Increase or decrease at least by minimal clinically important difference compared to baseline (4 for breast satisfaction, sexual well-being, and psychosocial well-beings in this study).

being and did not undergo chemotherapy during the cancer treatment. Additionally, patients with a low-not-restored trajectory in AUTOG (n = 28, 4.82%) were fewer than that in the IMPG (n = 110, 13.05%), but the clear trend of their trajectory was a persistent decline, rather than a gradual recovery over the 2-year follow-up. Results show these patients were characterized with the lowest PRO for all subscales during the evaluation period, undergoing radiation after reconstruction, being more of non-white ethnicity. The specific causes for the patient-reported continued decline of PWBC in patients with low-not-restored trajectory in AUTOG are multifactorial, and even autologous type may play a role here. Multiple studies have shown that patients undergoing pedicled transverse rectus abdominis myocutaneous (TRAM) flaps had poorer PWBC compared with patients undergoing implant reconstruction [13,29].

Regarding the significant predictors for PWBC trajectories, baseline PRO of PWBC was reasonably and strongly associated with all trajectories for both IMPG and AUTOG. Baseline PRO physical well-being of the abdomen and psychosocial well-being were consistently associated with all trajectories in AUTOG. We postulated that the PRO physical well-being abdomen was much more relevant for patients undergoing autologous reconstruction, however, the overall differences in psychosocial well-being between IMPG and AUTOG are not

demonstrable in existing literature [13]. Patients not undergoing radiation therapy were more likely to have a higher chance of developing high-restored or medium-low restored PWBC trajectories in IMPG, which is congruent with previous findings that the average score of physical well-being scale for radiotherapy patients treated with implant reconstruction was 7 to 9 points lower at all postoperative time points than these with no radiation after surgery [30].

In AUTOG, no axillary intervention was a significant predictor of the medium above PWBC trajectories, and SLNB was positively associated with medium-high restored and medium-low not restored trajectories, which is attributed to the fact that axillary management (no axillary intervention vs SLNB vs ALND) is always associated with the initial staging of the disease [31], and SLNB with lower arm morbidity clinically benefits patients on the quality of life over ALND [32]. Receiving chemotherapy was also consistently associated with medium above PWBC trajectories, nevertheless, prior research in women after immediate PMBR states chemotherapy had little impact on HRQOL [33].

More importantly, the results of the current study found intriguing differences between IMPG and AUTOG on HRQOL change. For instance, the current study revealed that patients with medium-low restored trajectories and axillary intervention of SLNB were more likely to have improved breast satisfaction in IMPG, which may be due to the less morbidity of SLNB compared to ALND [34]. However, patients receiving therapeutic mastectomy were more likely to experience worsened breast satisfaction in IMPG, suggesting that prophylactic mastectomy offer more benefits in patients on the quality of life over therapeutic mastectomy [14].

## Clinical and research implications

First, the strong and significant relationship between baseline scores from Breast-Q subscales and HRQOL outcomes after PMBR highlights the effectiveness of PRO measurements in collecting actionable data with higher signal and granular information to inform decision-making and promote patient-centered care [35,36]. Its excellent performance and convenient usage maybe deserve more attention from clinicians and researchers. Second, much more attention should be paid to patients with the lowest initial level of PWBC and all the patients after surgery, particularly in the first 3 months. Timely and effective resilience-enhancing interventions to alleviate the side effect of reconstruction procedures may prioritize the urgent needs of patients maintaining a medium high-restored trajectory in IMPG or developing a low-not restored trajectory in AUTOG. Third, interventions to facilitate postoperative rehabilitation of PWBC may consider the influence of significant factors such as the timing of radiation, type of axillary, chemotherapy receiving, and baseline PRO scores on the patients' PWBC trajectories. Fourth, regular evaluation and timely treatment are warranted for patients with a higher risk of worsened HRQOL outcome following PMBR, including the patients with a low-not restored trajectory in both groups, based on these ascertained significant predictors. Fifth, patient-centered supportive care should be also provided to improve postoperative HRQOL.

## Limitations

Several limitations associated with this study warrant further discussion. First, the sample size of AUTOG is somewhat smaller compared to IMPG, leading to few patients existing for the category of some variables (e.g., nipple-sparing mastectomy). The proportion of whites (90.59%) outnumbered non-whites (8.71%). Future studies recruiting more patients with more racial diversity to enhance the generalizability of our findings and using medical diagnoses to validate our conclusions are particularly warranted. Second, the physical well-being of the abdomen as an associated health outcome has not been examined here due to the lack of suggested MCID. Third, we observed a mixed impact of smoke status on health outcomes,

probably due to fewer (1.76%) smokers included here which may have been biased by self-reporting smoking behaviors out of social desirability concerns [37]. Fourth, Growth mixture modeling may be considered in future studies to further investigate within- and between groups changes on this topic [19]. Fifth, adding assessment time points during extended follow-up duration in future studies may depict a larger picture of long-term changes in physical well-being and related health outcomes in cancer patients in future research.

## Conclusion

Using longitudinal data from 2-year follow-up patients after PMBR, we investigated heterogeneous patterns of PWBC change over time and observed significant differences between IMPG and AUTOG. The four trajectories developed were relatively flat in AUTOG as opposed to that in IMPG. Recovery trajectories had a significant influence on HRQOL outcomes. The significant clinical and socioeconomic predictors associated with breast satisfaction, and sexual, and psychosocial well-being change were identified. All these data-driven findings can contribute to the enhanced understanding of the long-term effect of different PMBR modalities on the PWBC recovery process, which may inform clinical treatment decision-making, guide innovation of rehabilitation interventions on physical function, and tailor patient-oriented postoperative care to enhance patients' satisfaction with HRQOL outcomes.

## Supporting information

**S1 File.**
(DOCX)

**S1 Data. Checklist.**
(CSV)

## Acknowledgments

We would like to thank Dr. Edwin G. Wilkins of the Department of Surgery at the University of Michigan for all his assistance in the data collection process and for his valuable feedback on this improved manuscript. We acknowledge the Mastectomy Reconstruction Outcomes Consortium site principal investigators for their contributions: Yoon S. Chun, MD (Brigham and Women's Hospital), Richard Greco, MD (Georgia Institute of Plastic Surgery), Troy A. Pittman, MD (Georgetown University), Mark W. Clemens, MD (MD Anderson Cancer Center), John Kim, MD (Northwestern University), Daniel Sherick, MD (Saint Joseph Mercy Hospital), Gayle Gordillo, MD, Ed (The Ohio State University), Ed Buchel, MD (University of Manitoba), and Nancy Van Laeken, MD (University of British Columbia). We also thank all the patients who participated in the Mastectomy Reconstruction Outcome Consortium study and made this work possible.

## Author Contributions

**Conceptualization:** Cai Xu, Peiyi Lu, André Pfob, Andrea L. Pusic, Jennifer B. Hamill, Chris Sidey-Gibbons.

**Data curation:** André Pfob, Jennifer B. Hamill, Chris Sidey-Gibbons.

**Formal analysis:** Cai Xu, André Pfob, Chris Sidey-Gibbons.

**Funding acquisition:** Andrea L. Pusic, Chris Sidey-Gibbons.

**Investigation:** Peiyi Lu, André Pfob, Andrea L. Pusic, Jennifer B. Hamill, Chris Sidey-Gibbons.

**Methodology:** Cai Xu, André Pfob, Andrea L. Pusic, Jennifer B. Hamill, Chris Sidey-Gibbons.

**Project administration:** Cai Xu, Chris Sidey-Gibbons.

**Resources:** Peiyi Lu, André Pfob, Andrea L. Pusic, Jennifer B. Hamill, Chris Sidey-Gibbons.

**Software:** Cai Xu, André Pfob.

**Supervision:** Andrea L. Pusic, Jennifer B. Hamill, Chris Sidey-Gibbons.

**Validation:** Cai Xu, Peiyi Lu.

**Visualization:** Cai Xu, Peiyi Lu, Andrea L. Pusic, Chris Sidey-Gibbons.

**Writing – original draft:** Cai Xu, Chris Sidey-Gibbons.

**Writing – review & editing:** Cai Xu, Peiyi Lu, André Pfob, Andrea L. Pusic, Jennifer B. Hamill, Chris Sidey-Gibbons.

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
