## [Decision Letter · Decision Letter 0]

26 May 2023

PONE-D-23-11116Physical well-being recovery trajectories by reconstruction modality in women undergoing mastectomy and breast reconstruction: significant predictors and health-related quality of life outcomesPLOS ONE

Dear Dr. Xu,

Thank you for submitting your manuscript to PLOS ONE. After careful consideration, we feel that it has merit but does not fully meet PLOS ONE’s publication criteria as it currently stands. Therefore, we invite you to submit a revised version of the manuscript that addresses the points raised during the review process.

We look forward to receiving your revised manuscript.

Kind regards,

Shimpei Miyamoto

Academic Editor

PLOS ONE

“Supported by National Cancer Institute Grant No. R01 CA152192 and in part by National Cancer Institute Support Grant No. P30 CA008748.”

“Cai Xu

No relationship to disclose.

Peiyi Lu

No relationship to disclose.

André Pfob

No relationship to disclose.

Andrea L. Pusic

Patents, Royalties, Other Intellectual Property: I am a codeveloper of BREAST-Q and receive royalty payments when it is used in for-profit industry-sponsored trials.

Jennifer B Hamill

No relationship to disclose.

Chris Sidey-Gibbons

No relationship to disclose.”

Reviewers' comments:

Reviewer's Responses to Questions

**Comments to the Author**

1. Is the manuscript technically sound, and do the data support the conclusions?

Reviewer #1: Yes

2. Has the statistical analysis been performed appropriately and rigorously? 

Reviewer #1: Yes

3. Have the authors made all data underlying the findings in their manuscript fully available?

Reviewer #1: Yes

4. Is the manuscript presented in an intelligible fashion and written in standard English?

Reviewer #1: Yes

5. Review Comments to the Author

Reviewer #1: This paper describes the longitudinal changes in physical well-being of the chest up to 2 years after breast reconstruction by surgical technique using latent class growth analysis (LCGA) and identify factors influencing this change.

The statistical analysis is well described, and the results are clearly presented.

The main research findings of this paper will be important for understanding of quality of life after breast reconstruction, contributing to improved patient care.

I would recommend it for acceptance after the minor points listed below.

１）Regarding Table 1, there are differences in the four PWBC trajectories between IMPG and AUTOG (e.g., IMPG includes 'medium high-not restored' but AUTOG does not). To facilitate the reader's visual understanding, it is suggested that this be corrected so that the class and graph color corresponds one-to-one.

２）The second line of the abstract states "in patients who underwent postmastectomy and breast reconstruction,". Would it be better to correct it to "postmastectomy breast reconstruction" or "mastectomy and breast reconstruction"?

３）"his/" is inappropriate because the study subjects are all women. (p.10 ; line10)

４）The grammar of "The physical well-being score was AUTOG is significantly better than that in IMPG[11,28]." should be corrected.(p.21 ; lines18-19)

I hope these comments will be helpful.

6. PLOS authors have the option to publish the peer review history of their article (what does this mean?). If published, this will include your full peer review and any attached files.

Reviewer #1: No

---

## [Author Response · Author response to Decision Letter 0]

9 Jun 2023

Responses to Reviewer1

Reviewer #1: This paper describes the longitudinal changes in physical well-being of the chest up to 2 years after breast reconstruction by surgical technique using latent class growth analysis (LCGA) and identify factors influencing this change.

The statistical analysis is well described, and the results are clearly presented.

The main research findings of this paper will be important for understanding of quality of life after breast reconstruction, contributing to improved patient care.

I would recommend it for acceptance after the minor points listed below.

Response:

Thank you for your valuable feedback and positive assessment of our paper. We appreciate your recognition of the clarity in our statistical analysis and presentation of results, as well as the significance of our research findings for enhancing quality of life after breast reconstruction and improving patient care.

Your detailed suggestions have greatly contributed to refining and strengthening our paper. We are fully committed to upholding the highest research standards and greatly value your expertise in helping us achieve this goal.

We have incorporated the corresponding changes into the manuscript's main body, highlighted in red, and provided a detailed explanation for each change below.

1)Regarding Table 1, there are differences in the four PWBC trajectories between IMPG and AUTOG (e.g., IMPG includes 'medium high-not restored' but AUTOG does not). To facilitate the reader's visual understanding, it is suggested that this be corrected so that the class and graph color corresponds one-to-one.

Response: 

Thanks for this good suggestion.

We have adjusted the colors in both graphs to establish a one-to-one correspondence between the classes and the graphs, to facilitate the reader's visual understanding. See these new plots below.

2)The second line of the abstract states "in patients who underwent postmastectomy and breast reconstruction,". Would it be better to correct it to "postmastectomy breast reconstruction" or "mastectomy and breast reconstruction"?

Response:

Thanks for pointing out these details.

I took your advice and corrected this sentence to read as “in patients who underwent postmastectomy and breast reconstruction” in Abstract.

Actually in main body of context, it keeps using “postmastectomy breast reconstruction”, or PMBR for short.

PMBR=postmastectomy breast reconstruction

3)"his/" is inappropriate because the study subjects are all women. (p.10; line10)

Response: 

We agree with the reviewer’ comments and thanks for this interesting suggestion.

We removed the word” his/”, our patients here are all women. See page 10 for details.

“each included patient will be assigned a new class membership based on their respective posterior class probabilities to represent his/her growth patterns over time, and their respective new class membership attribution will not change at all the assessment time points.”

4)The grammar of "The physical well-being score was AUTOG is significantly better than that in IMPG[11,28]." should be corrected. (p.21; lines18-19)

Response: Thanks for your helpful suggestions.

I took you advice and changed this sentence in p.21 to this:

“The physical well-being score in AUTOG is significantly better than that in IMPG[11,28].”

Responses to Journal requirements:

Response: Yes. Our manuscript meets PLOS ONE's style requirements.

Response: Yes, we added a subsection ‘Inclusivity in global research’ to our Methods section and adding the following sentence: “Additional information regarding the ethical, cultural, and scientific considerations specific to inclusivity in global research is included in the Supporting Information (S2 Checklist)” in p.10.

Of note, the study was originally IRB approved in 2012 which is before single IRB requirements. Each individual center managed its own regulatory process with support from us.

We completed this questionnaire and uploaded it as Supporting Information when we resubmit our manuscript.

“Supported by National Cancer Institute Grant No. R01 CA152192 and in part by National Cancer Institute Support Grant No. P30 CA008748.”

Response: Role of Funder statement

“Cai Xu

No relationship to disclose.

Peiyi Lu

No relationship to disclose.

André Pfob

No relationship to disclose.

Andrea L. Pusic

Patents, Royalties, Other Intellectual Property: I am a codeveloper of BREAST-Q and receive royalty payments when it is used in for-profit industry-sponsored trials.

Jennifer B Hamill

No relationship to disclose.

Chris Sidey-Gibbons

No relationship to disclose.”

Response: Competing Interests statement

This does not alter our adherence to PLOS ONE policies on sharing data and materials.

Response:

All references listed in this manuscript are complete and correct. No new references were added during the revision process.

---

## [Editor Report · Decision Letter 1]

13 Jul 2023

Physical well-being recovery trajectories by reconstruction modality in women undergoing mastectomy and breast reconstruction: significant predictors and health-related quality of life outcomes

PONE-D-23-11116R1

Dear Dr. Xu,

We’re pleased to inform you that your manuscript has been judged scientifically suitable for publication and will be formally accepted for publication once it meets all outstanding technical requirements.

Kind regards,

Shimpei Miyamoto

Academic Editor

PLOS ONE
---

## [Editor Report · Acceptance letter]

19 Jul 2023

PONE-D-23-11116R1 

Physical well-being recovery trajectories by reconstruction modality in women undergoing mastectomy and breast reconstruction: significant predictors and health-related quality of life outcomes 

Dear Dr. Xu:

I'm pleased to inform you that your manuscript has been deemed suitable for publication in PLOS ONE. Congratulations! Your manuscript is now with our production department. 

Kind regards, 

on behalf of

Dr. Shimpei Miyamoto 

Academic Editor

PLOS ONE